# A Novel One-Step Reactive Extrusion Process for High-Performance Rigid Crosslinked PVC Composite Fabrication Using Triazine Crosslinking Agent@Melamine-Formaldehyde Microcapsules

**DOI:** 10.3390/ma16134600

**Published:** 2023-06-26

**Authors:** Jinshun Zhao, Chun Li, Jiayang Sui, Shuai Jiang, Weizhen Zhao, Shihao Zhang, Rong Wu, Jintong Li, Xuhuang Chen

**Affiliations:** 1Hubei Provincial Key Laboratory of Green Materials for Light Industry, School of Materials and Chemical Engineering, Hubei University of Technology, Wuhan 430068, China; zhaojinxun2021@ipe.ac.cn (J.Z.); zsh@hbut.edu.cn (S.Z.); 2Beijing Key Laboratory of Ionic Liquids Clean Process, Institute of Process Engineering, Chinese Academy of Sciences, Beijing 100190, China; lichun@ipe.ac.cn (C.L.); jysui@ipe.ac.cn (J.S.); sjiang@ipe.ac.cn (S.J.); wurong2021@ipe.ac.cn (R.W.); lijintong2021@ipe.ac.cn (J.L.); 3Longzihu New Energy Laboratory, Zhengzhou Institute of Emerging Industrial Technology, Henan University, Zhengzhou 450046, China

**Keywords:** crosslinking, extrusion, microcapsules, nanocomposites

## Abstract

In this work, we propose, for the first time, a simple, fast, and efficient strategy to fabricate high-performance rigid crosslinked PVC composites by continuous extrusion. This strategy improves the poor processing fluidity of composites and solves the impossibility of conducting extrusion in one step via using microcapsule-type crosslinking agents prepared by in situ polymerization to co-extrude with PVC blends. The results demonstrate that the PVC/microcapsule composites were successfully prepared. Within the studied parameters, the properties of crosslinked PVC gradually increased with the addition of microcapsules, and its Vicat softening temperature increased from 79.3 °C to 86.2 °C compared with pure PVC. This study shows the possibility for the industrial scale-up of the extrusion process for rigid crosslinked PVC.

## 1. Introduction

Poly(vinyl chloride) (PVC) is one of the most widely used materials in plastic engineering owing to its excellent properties as well as its low ingredient costs [1]. It is broadly applied in the fields of plastic wood, wind power, aerospace, medical equipment, and conveying pipes [2] due to its valuable mechanical properties and dimensional stability at room temperature. However, the utilization of PVC is always limited by its low Vicat softening temperature. When the external temperature reaches a certain value, its mechanical properties will drop greatly [3]. Therefore, determining how to improve the heat resistance of PVC has been a critical issue that has attracted significant interest from industry and academia.

The network structures between PVC chains that use crosslinking agents [4,5,6] limit interchain slippage and increase the stiffness of chain segments, which can improve heat resistance and mechanical properties [7]. Traditional processing directly compounds PVC blends with crosslinking agents. When the processing temperature is higher than the crosslinking temperature, the crosslinking reaction occurs. Normally, two-step compression moulding is used to fabricate crosslinked PVC, as shown in Appendix A. However, extrusion is also one of the largest-scale technologies in the industry [8,9] with outputs of more than 10 tons/h [10]. When extrusion is used to fabricate crosslinked PVC, higher temperatures make premature crosslinking unavoidable [11]. Crosslinked structures in the extruder make the relative movement of the chain molecules difficult, resulting in poor processing fluidity of PVC composites and an inability to prepare crosslinked PVC. Therefore, it is important to improve the processing fluidity to achieve continuous extrusion of crosslinked PVC.

Polymeric microcapsules (MCs), which utilize natural or synthetic polymeric materials to encapsulate solids, liquids, or gases in small containers with a hermetic layer, are considered a potential technology to provide a controlled release system for active ingredients [12,13,14,15,16,17,18,19]. Regarding the isolation of MCs, it has been reported that the encapsulation of *n*-dodecanol via MC technology is widely used in phase change materials [20], which indicates that MCs are effective for protecting core materials from external environmental conditions. The sustained release effect of MCs has been investigated in acid-responsive drug carrier MCs, allowing the drug to be effective for a long time [21], which confirms that MCs exhibit excellent sustained release ability. In addition, studies on the controlled release of MCs have shown that coating with self-healing MCs restrains metal corrosion after coating breakdown [22], which shows that MCs can successfully release corrosion inhibitors under the prescribed level of external mechanical forces. Therefore, encapsulating the crosslinking agents into MCs can avoid premature crosslinking and improve processing fluidity and release the crosslinking agents during subsequent processing to complete the crosslinking reaction, providing an effective means of regulating fluidity.

For the selection of the encapsulant, the polymer shell must be able to withstand the extrusion conditions of PVC (160 to 190 °C) [23,24]. Several heat-resistant polymers with good mechanical stability have been publicly reported [25,26]. Melamine-formaldehyde (MF) has been widely used as an encapsulant in studies on microencapsulation technology [27,28,29]. Inspired by this, here, we propose a strategy of applying MC technology to crosslinking extrusion: 2-dibutylamino-4,6-dimethylmercapto-1,3,5-triazine (DB) is used as the core incorporated into MCs with MF as the shell, and microcapsule-type crosslinking agents (DB@MF) are synthesized by in situ polymerization [30]. During extrusion, the MF shell is destroyed by the screw shear, and crosslinking agents slowly escape through the cracks of the shell to react with PVC. By utilizing MC technology to regulate the processing fluidity and adjusting the process parameters, smooth extrusion can be achieved. The properties of PVC/DB@MF composites were determined through various tests, including rheological tests, dynamic mechanical analysis, tensile tests, flexural tests, Vicat softening temperature measurements, and thermogravimetric analysis. Additionally, the surface morphologies of MCs and the fracture surfaces of the composites were observed by scanning electron microscopy.

## 2. Materials and Methods

### 2.1. Materials

Melamine, formaldehyde, acetic acid, magnesium oxide, and sodium dodecyl benzene sulfonate (SDBS) were purchased from Sinopharm Chemical Reagent Limited Corporation (Shanghai, China). 2-Dibutylamino-4,6-dimethylmercapto-1,3,5-triazine (DB) was purchased from Guangzhou Qiazhan Chemical Products Limited Corporation (Guangzhou, China). Sodium hydroxide was purchased from the Damao Chemical Reagent Factory (Tianjin, China). For the PVC (SG-5, average degree of polymerization: 1100-1000) was supplied by Xinjiang Tianye Limited Company (Xinjiang, China). Calcium-zinc stabilizer (PRJ5030G-9) was purchased from Nanjing Jinling Chemical Factory Limited Liability Company (Nanjing, China).

### 2.2. Preparation of DB@MF and PVC/DB@MF Composites

Melamine and formaldehyde were dissolved in a three-necked flask under stirring (Appendix A). The pH was subsequently adjusted to 8.5 using NaOH solution. Then, the mixture system was heated up to 60 °C and stirred for 1.5 h to produce a colorless, viscous prepolymer. Simultaneously, the suspension was prepared by SDBS surfactant and DB with ultrasonication. Next, the suspension and prepolymer were placed in the reaction flask at a temperature of 60 °C. To activate the reaction, acetic acid solution was added to change the pH value from 8.5 to 5.5. The solution was left to react and stirred for 2.5 h. After the completion of microencapsulation, the DB@MF was collected, washed, and dried to obtain the powder.

The PVC blends and DB@MF were placed in an electric thermostatic drying oven to remove the water. Then, they were added to the high-speed mixing machine and mixed for 30 min at 1000 rpm and 90 °C to obtain the PVC/DB@MF composites. The samples containing pure PVC and PVC/DB@MF composites combined in 100:6 and 100:10 weight ratios are referred to as 0 phr, 6 phr, and 10 phr, respectively.

### 2.3. Instrumentation

The mechanical response of DB@MF was studied by force deformation experiments with atomic force microscopy (AFM) (FastScan Bio, Bruker, Karlsruhe, Germany). The composition of DB@MF was characterized by Fourier transform infrared spectroscopy (FTIR) (Nicolet-380, Thermo Fisher Scientific, Waltham, MA, USA) and a Raman spectrometer (inVia, Renishaw, Wotton-under-Edge, UK). The water contact angle (WCA) of DB@MF was measured with a drop shape analysis system (DSA 100S, Krüss, Hamburg, Germany). The thermal degradation behavior of DB@MF was characterized by simultaneous thermogravimetry/differential thermal analyzers (TG/DTA) (DTG-60, Shimadzu, Kyoto, Japan). The ratio of elements was researched via X-ray photoelectron spectroscopy (XPS) (Escalab 250Xi, Thermo Fisher Scientific, Waltham, MA, USA) and an elemental analyzer (EA) (Vario EL cube, Elementar, Langenselbold, Germany). The morphology of DB@MF was examined by using scanning electron microscopy (SEM-EDX) (SU8020, Hitachi, Tokyo, Japan) and transmission electron microscopy (TEM-EDX) (JEM-2100F, JEOL, Tokyo, Japan). The particle size analysis of DB@MF was conducted with a Zeta potential analyzer (DelsaNano C, Beckman Coulter, Brea, CA, USA). The release of DB@MF was investigated by a torque rheometer (RM200A, Hapro, Harbin, China). The PVC sheet was characterized by dynamic thermodynamic analysis (DMA) (Q800, TA Instruments, New Castle, DE, USA). The tensile strength, flexural strength, and Vicat softening temperature of PVC composites were investigated by a universal materials tester (DZ-101-2KN, Da Zhong, Dongguan, China) and a Vicat softening temperature tester (XRW-300F, Jinhe, Chengde, China), respectively.

### 2.4. Core Content of DB@MF

The core content of the synthesized DB@MF was determined by the EA [31]. The fully dried MCs were covered with tin foil. Afterwards, the samples were placed in the instrument for full combustion, and the elemental proportions of each sample were obtained. The S element comes from the core material and its amount remains unchanged after combustion, which means that the content of MCs can be evaluated by the S element. The core content of microcapsules (*α*) was calculated by the following formula:(1)$$\alpha =\frac{Mr(\mathrm{DB})}{2\times Ar(S)}\times S_{\mathrm{EA}}\times 100\%$$
where *Mr* is the relative molecular mass, *Ar* is the relative atomic mass, and *S*_EA_ is the percentage of the S element in the EA, respectively.

## 3. Results

### 3.1. Characterization of Microcapsules

Figure 1 shows that DB@MFs were prepared by in situ polymerization. Melamine reacts with formaldehyde through a nucleophilic addition reaction and forms hydroxymethyl melamine under alkaline conditions. At the same time, DB is added to the dispersant solutions, and ultrasonic treatment is used to form the DB suspension. By adding hydroxymethyl melamine solution to the suspension, the polycondensation of hydroxymethyl melamine can be carried out to build the formation of -O- or -CH_2_- bridge bonds under acidic conditions. With an increasing molecular weight, the water-solubility of polycondensation decreases. Eventually, an insoluble structure encapsulates the DB by forming a precipitate layer on the surface. In this reaction, the hydroxymethyl melamine has a positive charge via the electron-withdrawal of the hydroxymethyl groups under acidic conditions [32,33], thus creating an electrostatic interaction with the anionic surfactants and precipitating on the DB surface. Next, the chemical composition, morphologies, and mechanical properties of DB@MF were characterized.

The FT-IR spectra of MF, DB, and DB@MF are shown in Figure 2a. The triazine bands of MF and DB@MF were observed at 1565 cm^−1^, 1505 cm^−1^, and 812 cm^−1^ and were attributed to the stretching vibrations of C=N and C-N and the bending vibration of the triazine ring in melamine, respectively [34]. The stretching vibration of C=N in DB appeared at 1589 cm^−1^, which was different from that of DB@MF and MF. As the shell merely precipitated on the core material surface without a chemical reaction, the characteristic peak of the core materials remained unchanged. However, the characteristic peak of C=N in DB disappeared in DB@MF. In addition, the stretching vibration band C-O of the methylene ether linkage was observed at 1165 cm^−1^ [35] and was found in both MF and DB@MF. The above phenomenon indicates that the core material was successfully microencapsulated. Figure 2b shows the Raman spectra of MF, DB, and DB@MF. The strong peak at 976 cm^−1^ and the shoulder at 899 cm^−1^ in MF are typical of triazine ring breathing in melamine [36] and the C-O-C stretch in the ether bridge [37]. Additionally, the peak at 899 cm^−1^ appeared on DB@MF, which means that MF successfully formed sediments on the DB surface. In addition, when DB was encapsulated, a group of characteristic peaks at 891 cm^−1^ belonging to -C-SH [38] disappeared on DB@MF owing to the wrapping effect of the MF shell. Based on the analysis, the spectrum revealed only the well-defined absorption peaks of MF without obvious characteristic bands of DB. This indicates that DB particles were well-microencapsulated.

The WCA is an important parameter for evaluating the hydrophilicity and surface free energy of materials. As the WCA decreases, the material becomes more hydrophilic, and the surface free energy increases. Figure 3 shows that pure DB had a low surface free energy with a WCA of 113.84° (Figure 3a), indicating that DB has a hydrophobic surface. After pure DB was encapsulated by MF, it had a strong surface free energy due to the formation of hydroxymethyl melamine groups with powerful polarity. In this case, the WCA of DB@MF reached 61.94° and was similar to that of MF (Figure 3b,c), and the encapsulated DB turned from a hydrophobic surface into a hydrophilic surface. The above results indicate the presence of a well-covered MF coating on the surface of DB.

In addition, the thermal stability analysis was also performed. The weight loss curves of MF, DB, and DB@MF are shown in Figure 4a,b. It was found that pure DB had three weight loss stages. The stage with a maximum weight loss rate at 234 °C was in the range of 200–410 °C, which was caused by thermal degradation. There were four stages characterized in MF by different weight loss rates. As the temperature reached 200–420 °C, the maximum weight loss rate occurred at 381 °C. However, the maximum weight loss rate of DB@MF occurred at 320 °C, which was significantly later than that of the core materials. From what is stated above, MC technology significantly improved the thermal stability of the core, indicating that the dense MF layer had an inhibitory effect on decomposition. In addition, the *T*_10%_ of DB@MF (Table 1) was greater than the processing temperature of PVC, signifying that DB@MF did not undergo decomposition and release crosslinking agents at the processing temperature.

To quantitatively measure the elemental composition of DB@MF with different shell/core ratios, XPS and EA were used (Appendix A). Figure 5a shows the scanning XPS spectra obtained from DB, DB@MF-1, DB@MF-2, and DB@MF-3. The peaks at 162 eV and 398 eV were assigned to S 2p and N 1s, respectively. The XPS results show that with the addition of MF, the S element became weaker, which was attributed to the better deposition of MF on the DB surface. The MF chemical structure was expected from the N 1s spectra in Figure 5c. Their respective binding energies were 398.2 eV (-C-N=C), 399.2 eV (-NH-), and 400.0 eV (-N-CO), where -C-N=C belongs to the nitrogen of the melamine ring structure, -NH- was attributed to the nitrogen of the amino groups, and -N-CO was added to nitrogen in urea. This is in agreement with the previous results, indicating the successful synthesis of MF resin [39]. In addition, the peak in the S 2p region at 162.1 eV (-SH) [40] in Figure 5d was identified as sulfur in the thiols, indicating the presence of the crosslinking agents. However, as the addition of MF increased, the peak gradually disappeared, indicating that thiols were not detected within DB@MF. This indicates that the microcapsule structure of DB@MF was integrated, and the core content (Appendix A) of DB@MF was obtained by Figure 5b and Formula (1).

Additionally, in order to investigate the possible impact of the microcapsule morphology on subsequent experiments, SEM analysis was performed on all samples, as shown in Figure 6. Figure 6a represents pure DB and shows agglomeration between particles. Due to the lower amount of MF added, DB@MF-1 could not separate the nanoparticles of DB completely, resulting in the agglomeration of crosslinking agents. Additionally, Figure 6b shows the formation of DB@MF-1, consisting of partially agglomerated DB and microcapsules. After modification by MF encapsulation, the microscopy images demonstrate the successful synthesis of microcapsules of uniform size. The size of the capsules is generally determined by emulsification technology, with particle size being reduced by vigorous emulsion or ultrasonic techniques, and shell thickness being determined by polymer solution concentration. With different reaction conditions (Appendix A), the size range of these capsules is approximately 190–1300 nm (Figure 6 and Appendix A). Due to agglomeration, the particle size of DB@MF-1 is significantly larger than those of DB@MF-2 and DB@MF-3, with the particle size distribution being consistent with the SEM results shown in Appendix A. The TEM-EDX mapping analysis of DB@MF-3 shows that the sulfur element belonging to DB is inside the MCs (Figure 6e). In addition, Figure 6d demonstrates that DB@MF has a well-formed core-shell structure. The results presented above show that MF resin demonstrate that an effective and thick encapsulation layer was convincingly formed on the DB surface.

To investigate the mechanical properties of microcapsules, the mechanical properties of DB@MF with different shell/core ratios were measured by AFM (Appendix A) [27,41]. DB@MF-1 was not fully encapsulated, and its mechanical properties belonged to the encapsulated crosslinking agents. DB@MF-3, with the largest shell/core ratio, had the best mechanical properties, which was attributed to its thick coating composed of MF with a high modulus (Appendix A). DB@MF-2 had the poorest mechanical properties, and the addition of MF did not exhibit a positive correlation with the Young’s modulus of DB@MF. Combined with the SEM results, DB@MF-2 was completely encapsulated. The poorer mechanical properties could infer that less MF shell was precipitated on the DB surface due to the reduction of shell/core ratios. Therefore, for the fully encapsulated DB@MF-2 and DB@MF-3, it could be predicted that DB@MF-3 will be released later than DB@MF-2.

### 3.2. Characterization of PVC Composites

To further investigate the effect of DB@MF on the performance of PVC blends, PVC/DB@MF composites were compounded by a torque rheometer at 180 °C, and their performance was verified using DMA. Appendix A shows the formulation with DB@MF incorporated at varying amounts.

The increasing rate of torque is closely associated with the vulcanization rate, and the equilibrium torque value is closely linked to the degree of crosslinking in the PVC [42]. Figure 7a shows the variation in the torque rheological curves with different shell/core ratios of DB@MF. DB@MF-1 was not fully encapsulated, and the crosslinking agents had a poor, delayed effect. The chain of PVC completed crosslinking early, and the degradation of PVC occurred, resulting in a continuously decreasing torque during processing. As the shell/core ratios between DB@MF-2 and DB@MF-3 increased, the vulcanization rate and equilibrium torque value decreased, indicating that the release of crosslinking agents gradually became slower, and the degree of crosslinking gradually decreased. Therefore, the release of crosslinking agents was successfully delayed, and DB@MF-3, with the best mechanical properties, had the best slow-release effect. It is well-known that higher storage modulus values are obtained for thermosets with a higher degree of crosslinking [43]. The storage modulus of PVC gradually grew due to the newly created and tightly bound crosslinking bonds between the chains. The decreasing storage modulus in Figure 7b indicates that DB@MF with increasing shell/core ratios had a slower release of crosslinking agents, and the degree of crosslinking gradually decreased.

From Figure 7c,e, with the increasing addition of DB@MF-3 and screw speeds, the processing torque of PVC/DB@MF composites gradually increased, indicating that processing parameters can control the release rate of crosslinking agents. The storage modulus and tan delta demonstrated an arrangement trend, with high values of DB@MF-3 shown in Figure 7d. The tan delta had a peak, indicating that the PVC composites had good compatibility. In addition, a higher tan delta indicates a higher *T*_g_, and the *T*_g_ of the composites exhibited a positive correlation with the amount of DB@MF added. The results show that, with an increasing temperature, PVC composites underwent softening, while the presence of crosslinked structures maintained a stable structure, resulting in enhanced melt flow resistance. This demonstrates that the crosslinked structure formed in the composites acts as a stable skeleton, preventing the relative motion of chain molecules and thereby increasing the *T*_g_ of the composites. Similarly, the curves shown in Figure 7f with a higher storage modulus and tan delta could also indicate the successful release of crosslinking agents. The above results show that MC technology can regulate the processing fluidity. The degree of crosslinking of PVC composites was negatively proportional to the increasing shell/core ratios and positively proportional to the screw speed and the addition of DB@MF.

Next, the formulation in Appendix A was tested for twin screw extrusion. With improved processing fluidity, crosslinked PVC was successfully prepared by continuous extrusion in Appendix A. Table 2 shows that, with the addition of DB@MF-3, the crosslinked structures formed between the PVC chains, and the structures prevented the shift, enhancing the thermal and mechanical properties of PVC. Compared to the 0 phr (pure PVC), the mechanical properties and heat resistance of crosslinked PVC gradually improved. The flexural strength of the 10 phr was 66.17 MPa, and the Vicat softening temperature increased up to 86.2 °C. The tensile strength of crosslinked PVC was 44.82 MPa, which is 141% higher than that of the 0 phr. However, the elongation at break in the tensile testing was reduced. The presence of the crosslinked structure in the composites led to an increase in stiffness, resulting in an enhanced tensile strength and flexural strength. However, the increase in stiffness was often accompanied by a decrease in toughness and a significant reduction in elongation at break. Twin screw extrusion testing findings supported the results obtained previously for DMA data and reflected the enhancement of the mechanical performance and thermal properties after crosslinking. In summary, the extrusion of crosslinked PVC was successfully achieved.

To elucidate the crosslinking behavior, a fractographic analysis was conducted. Figure 8a,b shows the fracture surface of PVC before and after crosslinking. The fracture surface of the 0 phr was smooth, and only part of the filler could be seen. The 10 phr had a rougher fracture surface, and crosslinked structures could be seen. The above phenomenon indicates a strong bond between PVC and the crosslinked structure. During stretching, the crosslinked structure tried to resist the deformation and broke away from the PVC matrix, leaving a rough fracture surface. Then, the 10 phr was analyzed by using SEM-EDX, and it could be seen that there were S elements belonging to DB in the crosslinked structure shown in Figure 8c,d, indicating that crosslinking agents were successfully released to react with the PVC, and the presence of crosslinked structures was found in the composite. In addition, the decomposition of PVC after crosslinking was improved (Appendix A), demonstrating that the crosslinked structure can improve the decomposition of PVC.

To further confirm the successful release of crosslinking agents during processing, the PVC/DB@MF composites were analyzed by SEM-EDX before and after processing, as shown in Appendix A. As the percentage of TiO_2_ in the composites remained constant, the S/Ti ratio was used to determine the content of S. As shown in Table 3, the ratio of S/Ti increased after processing, which indicates that the content of S elements in the composites increased, proving that the crosslinking agents were successfully released after processing. The heat weight loss test of DB@MF, shown in Figure 4, indicated that the microcapsules had good thermal stability. DB@MF did not undergo degradation or release DB, even at processing temperatures, ruling out the possibility of the temperature influencing the release. The results obtained demonstrate that DB@MF released DB through a mechanical response to processing shear.

## 4. Conclusions

A novel strategy was proposed, wherein the extrusion of crosslinked rigid PVC was achieved through the incorporation of microcapsules. In this study, DB@MFs were prepared by in situ polymerization and mixed with PVC blends to improve the processing fluidity by delaying crosslinking. The release behavior of DB@MF had a mechanical response, which was investigated by TGA as well as SEM-EDX. As the shell/core ratios of DB@MF increased, its mechanical properties gradually enhanced, and the retardation effect became better. After DB@MF was mixed with PVC in the extruder, the mechanical properties of the composites were improved and positively correlated with the added amount of DB@MF. The release of crosslinking agents was controlled by the processing parameters, such as the microcapsule shell/core ratios, feeding rate, and screw speed. The PVC composites produced by the extrusion process showed remarkable mechanical improvements, heat resistance enhancement, and excellent productivity, resulting in high competitiveness as high-value-added materials for many applications. In addition, this processing method, which can manufacture crosslinked polymers in one step, may play an important role in polymer processing and deserves further investigation.

## Figures and Tables

**Figure 1 materials-16-04600-f001:**
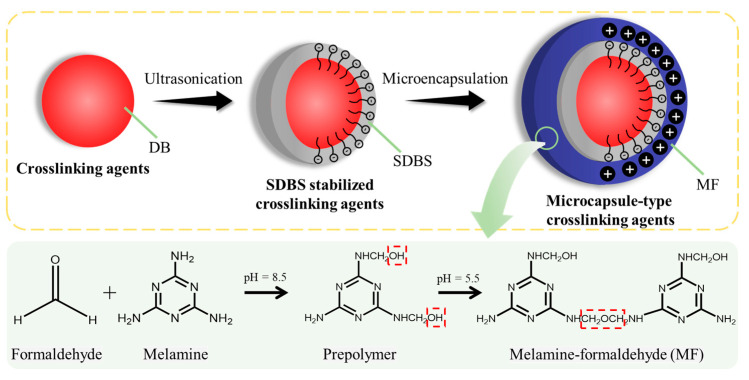
Schematic synthesis of microcapsule-type crosslinking agents.

**Figure 2 materials-16-04600-f002:**
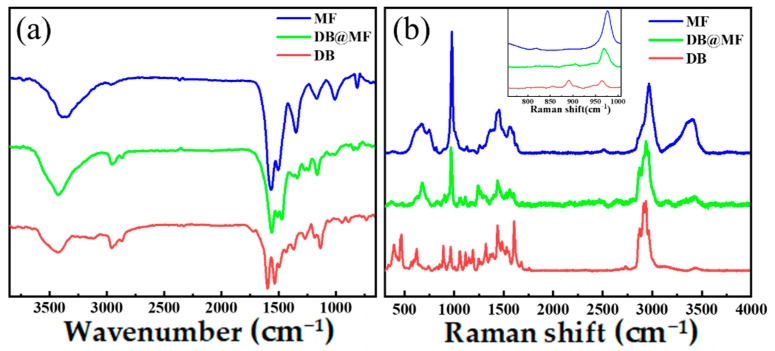
(**a**) FT–IR spectra and (**b**) Raman spectra of MF, DB@MF, and DB.

**Figure 3 materials-16-04600-f003:**
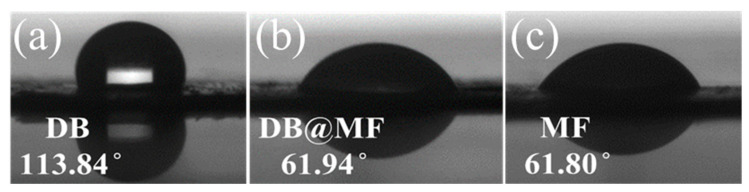
Contact angle analysis of (**a**) DB, (**b**) DB@MF, and (**c**) MF.

**Figure 4 materials-16-04600-f004:**
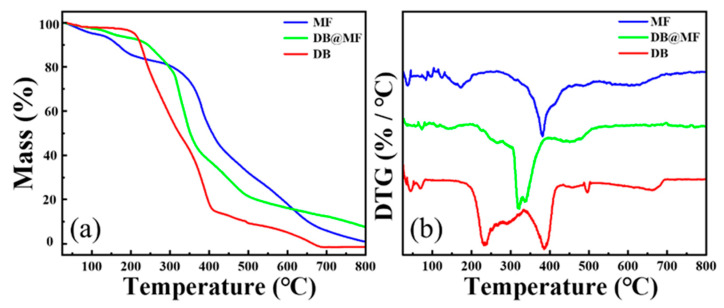
(**a**) Heat weight loss curves and (**b**) DTG curves of MF, DB@MF, and DB.

**Figure 5 materials-16-04600-f005:**
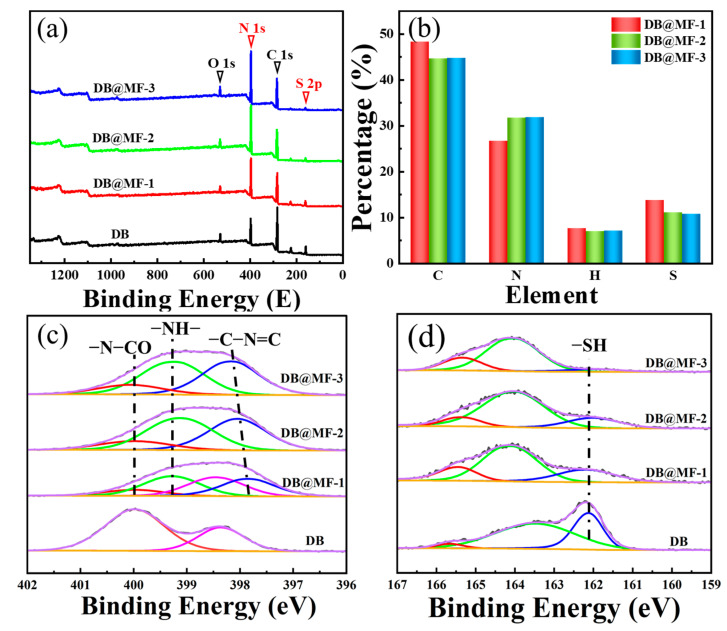
(**a**) XPS spectra of DB and DB@MF with different shell/core ratios. (**b**) EA of DB@MF with different shell/core ratios. High-resolution XPS spectra of (**c**) N 1s and (**d**) S 2p.

**Figure 6 materials-16-04600-f006:**
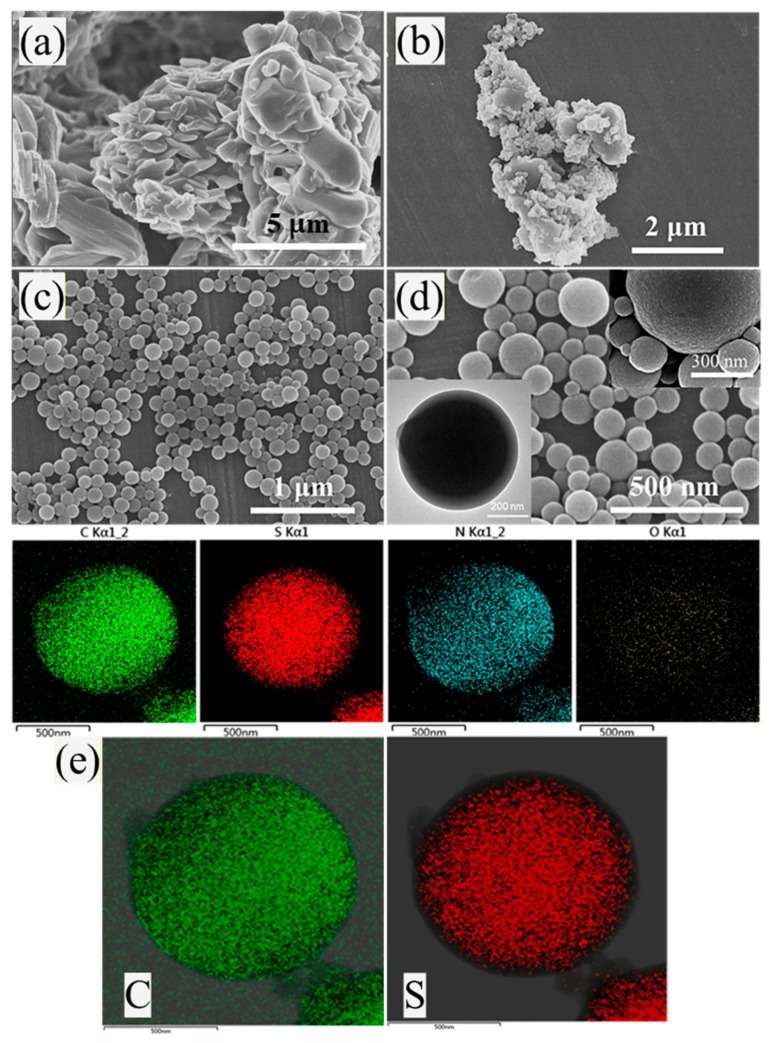
SEM micrographs of (**a**) DB, (**b**) DB@MF-1, (**c**) DB@MF-2 and (**d**) DB@MF-3. (**e**) TEM-EDX mapping of DB@MF-3.

**Figure 7 materials-16-04600-f007:**
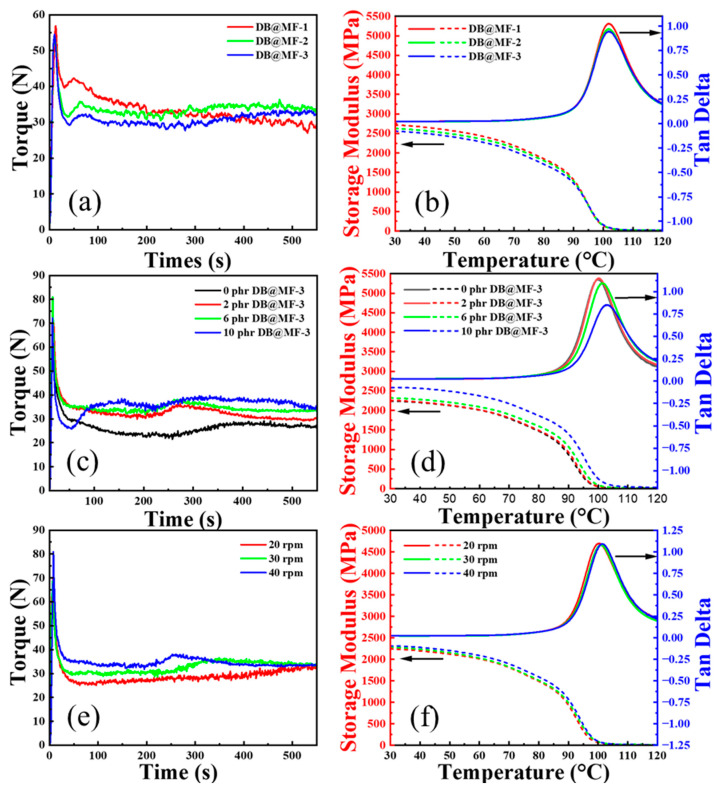
The torque diagram of PVC with (**a**) different shell/core ratios of DB@MF, (**c**) different additions of DB@MF–3, and (**e**) different processing speeds. The storage modulus and tan delta of PVC with (**b**) different shell/core ratios of DB@MF, (**d**) different additions of DB@MF–3, and (**f**) different processing speeds.

**Figure 8 materials-16-04600-f008:**
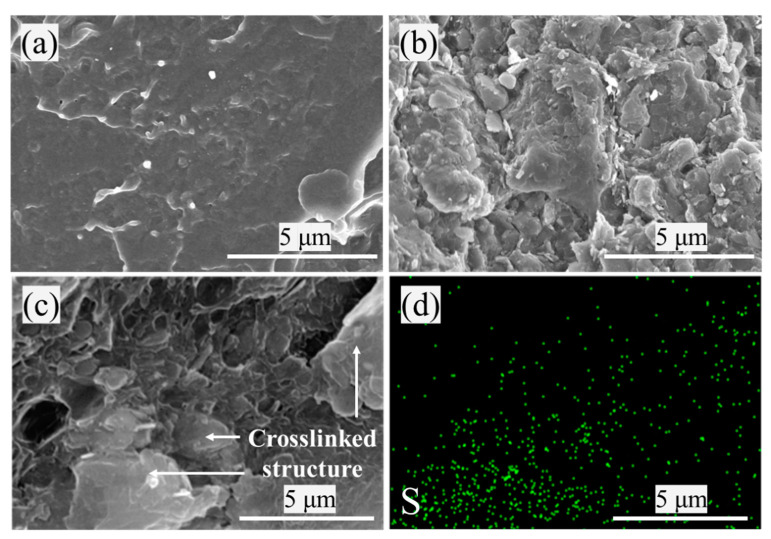
SEM micrographs of (**a**) 0 phr, (**b**) 10 phr. (**c**,**d**) SEM-EDX mapping of 10 phr.

**Table 1 materials-16-04600-t001:** Thermal analysis of data from DB, DB@MF, and MF.

Sample	*T*_10%_/°C	*T*_50%_/°C	*T*_max_/°C
DB	225	323	234/388
DB@MF	243	351	320
MF	166	408	381

**Table 2 materials-16-04600-t002:** Properties of PVC/DB@MF composites after extrusion.

Properties	0 phr	6 phr	10 phr	Growth Rate ^1^
Hardness	77.87	78.75	79.00	1.44%
Tensile strength (MPa)	31.85	39.82	44.82	40.69%
Flexural strength (Mpa)	51.92	54.02	66.17	27.43%
Elongation at break (%)	57.17	13.62	11.67	−79.59%
Gel fraction (%)	-	20.95	35.82	-
Vicat softening temperature (°C)	79.30	83.80	86.20	8.70%

^1^ Comparing the 0 phr and the 10 phr.

**Table 3 materials-16-04600-t003:** SEM-EDX of the PVC/DB@MF composites before and after processing.

Samples	S (%)	Ti (%)	S/Ti Ratio
PVC/DB@MF composites (unprocessed)	0.13	0.09	1.44
PVC/DB@MF composites (under torque rheometer)	0.51	0.21	2.43
PVC/DB@MF composites (under extrusion process)	0.44	0.18	2.44

## Data Availability

Data sharing is not applicable to this article.

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
