# Peer review of "A Novel One-Step Reactive Extrusion Process for High-Performance Rigid Crosslinked PVC Composite Fabrication Using Triazine Crosslinking Agent@Melamine-Formaldehyde Microcapsules"

_materials, 2023, doi:10.3390/ma16134600_

Round 1

Reviewer 1 Report

That is quite an exciting paper on the fabrication of PVP crosslinked composites. 

My comments are given below:

1. It would help if you considered rewriting the last two paragraphs of the introduction; there are several syntax mistakes.

2. How can you explain the agglomeration observed via SEM for DB@MF-1?

3.  You attributed the better mechanical properties of DB@MF-2 to a complete encapsulation. What does that mean? (makes no sense) You should explain it again.

4.  Further to the previous comment, how did you confirm the formation of the thinner shell layer in DB@MF-2? Based on this, you should expect the DB@MF-3 to form an even thinner layer and present the poorest mechanical properties out of the three structures.

Author Response

Dear Reviewer:

Thank you very much for your letter and the Reviewers’ comments on our paper submitted to the Materials (materials-2415778). The valuable comments and suggestions from reviewers not only helped us to improve the quality of our manuscript but also a valuable asset for our future research projects. On the basis of reviewers’ comments and suggestions, we have carefully revised our manuscript and necessary amendments and corrections have been made accordingly. Amended portions are marked with highlight in the revised manuscript and supporting information, we anticipate that after this critical revision our manuscript will meet the publication standard of your esteemed Materials.

The responses to reviewers’ comments are listed as following:

That is quite an exciting paper on the fabrication of PVC crosslinked composites.

My comments are given below:

  1. It would help if you considered rewriting the last two paragraphs of the introduction; there are several syntax mistakes.

Response: We gratefully appreciate for your comment. Based on your feedback, we have revised the introduction. This will help to ensure that the introduction section is more focused and highlights the key conclusions of our study in a clear and concise manner.

  1. How can you explain the agglomeration observed via SEM for DB@MF-1?

Response: Thanks for your suggestion. The SEM results showed that the pure DB (Figure 6a in manuscript) was nanoparticles with obvious agglomeration. Due to the less amount of MF addition, DB@MF-1 could not separate the nanoparticles of DB completely, resulting in the agglomeration of crosslinking agents. And Figure 1 (Figure 6b in manuscript) shows the formation of a mixture consisting of partially agglomerated DB and microcapsules.

Figure 1. Original image of DB@MF-1 in the manuscript.

  1. You attributed the better mechanical properties of DB@MF-2 to a complete encapsulation. What does that mean? (makes no sense) You should explain it again.

Response: Thank you for your suggestions. The original text: “DB@MF-2 had the lowest mechanical properties” is marked in red on page 8 of the manuscript. The Young's modulus of DB@MF in Table S1 was summarized in Figure 2. The results indicate that the addition of MF does not show a positive correlation with the Young's modulus of DB@MF. Among all samples, DB@MF-2 exhibits the lowest modulus. The modulus of DB@MF-1 is attributed to the unencapsulated DB, while that of DB@MF-3 is assigned to the increased addition of MF. Due to the lower amount addition of MF, the fully encapsulated DB@MF-2 has a less deposition of MF on DB surface, resulting in a smaller modulus than DB@MF-3 (large amount of MF deposited on DB surface) and DB@MF-1 (unencapsulated DB).

Figure 2. Young's modulus of DB@MF with different shell/core ratios.

  1. Further to the previous comment, how did you confirm the formation of the thinner shell layer in DB@MF-2? Based on this, you should expect the DB@MF-3 to form an even thinner layer and present the poorest mechanical properties out of the three structures.

Response: We gratefully appreciate for your comment. Based on your feedback, we have added the contact angle data to explain the thinner shell of DB@MF-2. The contact angles of DB@MF-1 and DB@MF-2 in Figure 3 are similar to those of pure DB (Figure 3a in manuscript), indicating that encapsulation fails to change the hydrophobicity of DB due to the thin MF shell on DB surface. And DB@MF-3 transforms the hydrophobicity of DB (Figure 3b in manuscript), indicating that DB@MF-3 has a thicker shell. In addition, microcapsules with excellent sustained release effects are desired to improve the processing fluidity of composites during the extrusion. A thinner shell would release the crosslinking agents too early and make premature crosslinking during processing, resulting in excessive torque and failure for extrusion. During the rheological process (Figure 7a in manuscript), DB@MF-2 with lower mechanical properties was less effective than DB@MF-3 in retarding the release, leading to premature crosslinking and higher torque. So, DB@MF-3 with better Young's modulus will show the best sustained release effect.

Figure 3. Contact angle analysis of DB@MF-1 and DB@MF-2.

Once again, thank you very much for your comments and advices.

On the basis of reviewers’ comments, we have critically revised our manuscript. The amendments in revised version of this manuscript do not distort the main theme and core idea of this manuscript. For your convenience, the amended portions of the paper have been highlighted in revised manuscript.

Finally, we would like to appreciate the editors and reviewers for their critical analysis, comments and suggestions which is a valuable asset and source of guidance for our future research efforts. Hopefully revised manuscript will meet the required standards of publication of your well-known journal.

Sincerely yours,

Dr. Weizhen Zhao, E-mail: wzzhao@ipe.ac.cn

May 26, 2023, Institute of Process Engineering, Chinese Academy of Sciences, China

Reviewer 2 Report

This paper describes an attempt to obtain crosslinked PVC by a microencapsulated crosslinking agents (MD) is melamine-formaldehyde resin shell. This is an interesting study, and it is recommended to be accepted for publication after some revision according to the comments below.

COMMENTS

1.

The authors should provide a drawing of the chemical structure of the applied crosslinker DB

2.

This study is submitted as a scientific study for Materials, which is a Q1 category journal. Therefore, providing only tradenames of substances is not sufficient. The authors should provide at least the chemical names and ratios of compounds of the applied heat stabilizer, which is mentioned only as heat stabilizer (PRJ5030G-9) in subsection 2.1.

3.

The authors should provide explanation why they add large amount of heat stabilizer of 8.54 phr in their formulation.

4.

The “a” and “2a” in Table S3 are absolutely unclear. Exact amounts should be provided and explained

5.

In case of sufficient crosslinking in the extruder, flow of the PVC resin should be blocked because the viscosity should dramatically increase with increasing the extent of crosslinking. However, this is not the case. The authors should explain.

6.

The most important characteristics of crosslinking, that is, the gel fraction determination is absolutely missing in this paper. The authors should provide the gel fraction of their crosslinked PVC. Otherwise, one can conclude that the authors have not reached crosslinking but only a blend (or composite) of the melamine-formaldehyde resin with PVC.

In this respect, it has to be noted that PVC degrades even in the presence of heat  stabilizers, and undergoes crosslinking by the reaction between the formed polyene sequences and oxidized polyenes as well. See e.g. the following references:

B. Ivan, B. Turcsanyi, T. Kelen, F. Tudos, Effect of Metal Stearate Stabilizers on the Thermal Degradation of PVC in Solution: The Reversible Blocking Mechanism of Stabilization. J. Vinyl Technol., 12, 126-135 (1990)

T. Kelen, B. Ivan, T. T. Nagy, B. Turcsanyi, F. Tudos, J. P. Kennedy, Reversible Crosslinking in PVC Thermal Degradation. Polym. Bull., 1, 79-84 (1978)

These should be considered, discussed and cited in the references by the authors in this manuscript.

7.

The TG curve in Figure S5, that is, the significant difference between PVC and the “crosslinked PVC”. The presented TG curves indicate that the amount of PVC is about 50% of the uncrosslinked PVC, and there is no any stabilization improvement as stated on page 11.  

Author Response

Dear Reviewer:

Thank you very much for your letter and the Reviewers’ comments on our paper submitted to the Materials (materials-2415778). The valuable comments and suggestions from reviewers not only helped us to improve the quality of our manuscript but also a valuable asset for our future research projects. On the basis of reviewers’ comments and suggestions, we have carefully revised our manuscript and necessary amendments and corrections have been made accordingly. Amended portions are marked with highlight in the revised manuscript and supporting information, we anticipate that after this critical revision our manuscript will meet the publication standard of your esteemed Materials.

The responses to reviewers’ comments are listed as following:

This paper describes an attempt to obtain crosslinked PVC by a microencapsulated crosslinking agents (MD) is melamine-formaldehyde resin shell. This is an interesting study, and it is recommended to be accepted for publication after some revision according to the comments below.

COMMENTS

1.The authors should provide a drawing of the chemical structure of the applied crosslinker DB.

Response: Thanks for your suggestion. The chemical structure of DB has been added to Figure S1c.

2.This study is submitted as a scientific study for Materials, which is a Q1 category journal. Therefore, providing only tradenames of substances is not sufficient. The authors should provide at least the chemical names and ratios of compounds of the applied heat stabilizer, which is mentioned only as heat stabilizer (PRJ5030G-9) in subsection 2.1.

Response: Thanks for your suggestion. The heat stabilizer (PRJ5030G-9), calcium-zinc stabilizer, is a commercial product. We apologize that the ratios of compounds are not available as the formula of the stabilizer is kept confidential by each manufacturer.

3.The authors should provide explanation why they add large amount of heat stabilizer of 8.54 phr in their formulation.

Response: We gratefully appreciate for your comment. In general, 5 phr of stabilizer was added to PVC composites during processing. And this paper is a crosslinked system, which requires a higher amount of stabilizer. So, 8.54 phr of stabilizer was added to PVC composites during extrusion.

4.The “a” and “2a” in Table S3 are absolutely unclear. Exact amounts should be provided and explained.

Response: Thanks for your suggestion. We apologize for the ambiguous explanation. For example, the core content of DB@MF-3 is 45.87%. When the crosslinking agents (a) is 10 phr, the actual amount of DB@MF-3 is 21.8 phr.

m(DB)=m(DB@MF-3)×α(DB@MF-3)=21.8×0.4587=10 phr

5.In case of sufficient crosslinking in the extruder, flow of the PVC resin should be blocked because the viscosity should dramatically increase with increasing the extent of crosslinking. However, this is not the case. The authors should explain.

Response: We gratefully appreciate for your comment. Due to the controlled release of the microcapsules, rigid PVC is crosslinked at the suitable place (compression section or homogenization section) by delayed crosslinking. In addition, the gel fraction of PVC is moderate, making it possible to achieve crosslinked extrusion.

6.The most important characteristics of crosslinking, that is, the gel fraction determination is absolutely missing in this paper. The authors should provide the gel fraction of their crosslinked PVC. Otherwise, one can conclude that the authors have not reached crosslinking but only a blend (or composite) of the melamine-formaldehyde resin with PVC.

In this respect, it has to be noted that PVC degrades even in the presence of heat stabilizers, and undergoes crosslinking by the reaction between the formed polyene sequences and oxidized polyenes as well. See e.g. the following references:

  1. Ivan, B. Turcsanyi, T. Kelen, F. Tudos, Effect of Metal Stearate Stabilizers on the Thermal Degradation of PVC in Solution: The Reversible Blocking Mechanism of Stabilization. J. Vinyl Technol., 12, 126-135 (1990)
  2. Kelen, B. Ivan, T. T. Nagy, B. Turcsanyi, F. Tudos, J. P. Kennedy, Reversible Crosslinking in PVC Thermal Degradation. Polym. Bull., 1, 79-84 (1978)

These should be considered, discussed and cited in the references by the authors in this manuscript.

Response: We gratefully appreciate for your comment. Based on your feedback, we supplemented the gel fraction with crosslinked PVC in Table 2.

7.The TG curve in Figure S5, that is, the significant difference between PVC and the “crosslinked PVC”. The presented TG curves indicate that the amount of PVC is about 50% of the uncrosslinked PVC, and there is no any stabilization improvement as stated on page 11.

Response: Thanks for your suggestion. The TG curve in Figure S5 is marked in red on page 12 of the manuscript. The thermal weight loss curve of the crosslinked composites at the 300-500°C stage is higher than that of the uncrosslinked PVC, indicating that the crosslinked structure improves the decomposition of the composites.

Once again, thank you very much for your comments and advices.

On the basis of reviewers’ comments, we have critically revised our manuscript. The amendments in revised version of this manuscript do not distort the main theme and core idea of this manuscript. For your convenience, the amended portions of the paper have been highlighted in revised manuscript.

Finally, we would like to appreciate the editors and reviewers for their critical analysis, comments and suggestions which is a valuable asset and source of guidance for our future research efforts. Hopefully revised manuscript will meet the required standards of publication of your well-known journal.

Sincerely yours,

Dr. Weizhen Zhao, E-mail: wzzhao@ipe.ac.cn

May 26, 2023, Institute of Process Engineering, Chinese Academy of Sciences, China
